# Legionnaires’ Disease in China Caused by *Legionella pneumophila* Corby

**DOI:** 10.3390/microorganisms11010204

**Published:** 2023-01-13

**Authors:** Pei-Xing Xu, Hong-Yu Ren, Ran Li, Xiao-Jing Jin, Zhan-Cheng Gao, Tian Qin

**Affiliations:** 1State Key Laboratory for Infectious Disease Prevention and Control, Collaborative Innovation Center for Diagnosis and Treatment of Infectious Diseases, National Institute for Communicable Disease Control and Prevention, Chinese Center for Disease Control and Prevention, Beijing 102206, China; 2Department of Respiratory Medicine, Peking University People’s Hospital, Beijing 100044, China

**Keywords:** *Legionella pneumophila*, antibiotic susceptibility, pathogenesis, comparative genome analysis

## Abstract

*Legionella pneumophila* is an intracellular pathogen causing pneumonia in humans. In February 2022, Legionnaires’ disease caused by *L. pneumophila* strain Corby in a patient with lung adenocarcinoma was identified for the first time in China. This paper includes the case report and phenotypic and genomic analysis of the Corby (ICDC) strain. Its biological characteristics were evaluated by antibiotic sensitivity testing and cytology experiments, and genomic analysis was performed to understand its genetic evolution. The patient’s clinical manifestations included cough, fever, pulmonary infiltration, and significantly decreased activity endurance. After empirical antimicrobial therapy, infection indicators decreased. The Corby (ICDC) strain was susceptible to nine antibiotics and exhibited strong intracellular proliferation ability. A phylogenetic tree showed that the Corby (ICDC) strain was closely related to the Corby strain, but under the pressure of a complex environment, its genome had undergone more rearrangement and inversion. The type IF CRISPR-Cas system was identified in its genome, and spacer analysis indicated that it had been invaded by several foreign plasmids, bacteria, and viruses during evolution. Legionnaires’ disease caused by *L. pneumophila* strain Corby may be ignored in China, and it is urgent to improve long-term monitoring and investigation of aquatic environments and patients with respiratory infections to prevent a large-scale outbreak of Legionnaires’ disease.

## 1. Introduction

Bacteria in the genus *Legionella* are water-borne microorganisms, mainly concentrated in natural and artificial aquatic environments in close contact with humans; for example, water from hot springs, air-conditioners, and cooling towers [1]. *Legionella* is spread through aerosols, enter human alveolar macrophages through the respiratory tract, and parasitize intracellularly [2], leading to Legionnaires’ disease, with fever and pneumonia as the main symptoms [3]. The epidemic mortality due to the disease can reach 30% [4]. People are cautious about the pollution of the aquatic environment in their daily lives; however, this pathogen continues to cause Legionnaires’ disease worldwide. To date, 70 species of *Legionella* have been identified (www.bacterio.net/legionella.html (accessed on 18 December 2022)), and more than 90% of the cases of Legionnaires’ disease are caused by *Legionella pneumophila* [5,6]. 

In February 2022, a patient with lung adenocarcinoma presented with Legionnaires’ disease caused by *L. pneumophila* strain Corby in Beijing. This is the first Corby strain to be isolated in China since the outbreak in England, and the second sequence to be publicly available globally. *Legionella pneumophila* strain Corby (serogroup 1, monoclonal antibody type Knoxville) is a highly virulent strain isolated from a human legionellosis case [7] and is a model strain used for the pathogenesis of *L. pneumophila* [8,9] and the comparative genomic analysis with Philadelphia, Paris, Lens [10]. In aerosol-infected guinea pigs, the strain multiplies very rapidly in the lung and spreads to the blood, liver, spleen, and kidney [7,11]. In the genome of *L. pneumophila* Corby, a new type of conjugation/type IVA secretion system (*trb/tra*) was identified and more T4ASSs are encoded by the *trb/tra* genes on the genomic islands Trb-1 and Trb-2. The islands Trb-1 and Trb-2 are integrated within the tRNA^Pro^ gene (*lpc2778*) and the tmRNA gene (*lpc0164*), respectively [12]. At present, few studies have reported that *L. pneumophila* Corby has infected humans to cause disease and few have sought to understand its genetic evolution and virulence through its genome and biological characteristics.

In this study, we assessed its antibiotic susceptibility to nine clinical routine drugs for the treatment of Legionnaires’ disease, mastered the intracellular proliferation ability in macrophages, and investigated a comparative genomic analysis to identify different features such as the CRISPR-Cas system and degree of invasion by foreign genes. Our goal was to reveal the pathogenicity, drug resistance and genomic evolution of *L. pneumophila* strain Corby by exploring its biological and genomic characteristics, report the first case of the Corby strain in China, and reveal its threat to public health.

## 2. Materials and Methods

### 2.1. Sample Information and Media

*Legionella pneumophila* strain Corby (ICDC (The new Corby strain we found from a patient sample this time, named *Legionella pneumophila* Corby (ICDC))) was isolated from the sputum of the patient. On 16 February 2022, the patient was admitted to the hospital for 21 days owing to cough and expectoration for 5 months and aggravated shortness of breath for 1.5 months. Peripheral blood, sputum, and bronchoalveolar lavage fluid (BALF) were collected for bacterial isolation. Clinical data of the patient were collected, including white blood cell count, percentage of neutrophils, C-reactive protein, Next-generation sequencing (NGS) data (peripheral blood, bronchoalveolar lavage fluid, and bronchial brushing specimens), and comprehensive computed tomography (CT) data. The data about antibiotic treatment and outcomes were recorded. *Legionella pneumophila* was isolated on glycine, vancomycin, polymyxin B, cycloheximide (GVPC) (Oxoid) agar plates and cultured on buffered charcoal–yeast extract (BCYE) agar plates. 

### 2.2. DNA Extraction and Identification of Legionella pneumophila Strain Corby

The DNA was extracted using a DNA extraction kit (QIAamp DNA Mini Kit, QIAGEN, Hilden, Germany) following the manufacturer’s instructions, and the DNA concentration and quality were assessed using a NanoPhotometer spectrophotometer. The quality-qualified DNA was analysed by whole-genome sequencing (WGS) at Novogene Co., Ltd. (Beijing, China).

### 2.3. Antimicrobial Susceptibility Testing

Antimicrobial susceptibility testing was performed according to EUCAST recommendations and manufacturers’ instructions [13]. The minimum inhibitory concentrations (MICs) of azithromycin, erythromycin, rifampicin, moxifloxacin, levofloxacin, ciprofloxacin, tigecycline, clindamycin, and doxycycline were determined using E test strips (LIOFILCHEM). *Legionella pneumophila* strain Corby (ICDC) grown on BCYE agar with α-ketoglutarate (BCYE-α) for 48 h were resuspended in phosphate-buffered saline (PBS), and the suspension was adjusted to 0.5 McFarland standard. Then, the surface of a BCYE-α plate was evenly swabbed with a sterile cotton swab dipped in bacterial solution, and a single-gradient strip was placed in the centre. After incubation at 35 °C for 48 h, the MIC was determined as the concentration indicated on the strip at the intersection of the inhibition zone and the gradient zone. *Legionella pneumophila* strain ATCC 33152 was used as a control in the susceptibility experiments. Results were interpreted according to the epidemiological cut-off (ECOFF) for resistance breakpoints identified by EUCAST guidelines. Each antibiotic susceptibility test was repeated three times.

### 2.4. Intracellular Growth Assay

Mouse macrophages J774 were routinely cultured in Dulbecco’s modified Eagle medium-high glucose (DMEM-H) medium (GIBCO), supplemented with 10% foetal bovine serum (FBS) at 37 °C with 5% CO_2_. 

*Legionella pneumophila* in the logarithmic growth phase was used to infect mouse macrophages J774. Suspensions of *L. pneumophila* strain Corby (ICDC) and *L. pneumophila* strain philadelphia1 (JR32) (approximately 1 × 10^8^ cells/mL) were diluted 10-fold in DMEM-H medium, and then J774 cells were infected at a multiplicity of infection of 100. Infected cells were incubated for 1.5 h at 37 °C in 5% CO_2_; then, the bacterial solution was discarded, and the cells were washed three times with 500 µL PBS to remove extracellular bacteria. Next, 0.5 mL DMEM-H medium containing 10% FBS was added to each well. To determine the number of proliferating bacteria in J774 cells, the intracellular and extracellular bacteria in each well were combined at 24 h intervals, and the number of proliferating bacteria was determined by plating the cell suspension on BCYE plates. Each experiment was repeated three times.

### 2.5. Bioinformatics

The assembled Corby (ICDC) sequences were uploaded to the RAST server (https://rast.nmpdr.org/rast.cgi, accessed on 8 September 2022) to annotate individual gene functions online, and then the annotation file was uploaded to Proksee (https://proksee.ca/projects/new, accessed on 8 September 2022) to acquire a genome visualization for the *L. pneumophila* strain Corby (ICDC) chromosome. Cluster analysis of the Corby (ICDC) and reference strain (serogroup 1) protein sequences was performed using cd-hit software to identify single-copy core genes, and they were aligned using MUSCLE. Phylogenetic tree was constructed using Treebest software using the neighbour-joining (NJ) method. Colinear alignment analysis of the two isolated Corby genomes was performed using MAUVE.

The clusters of regularly interspaced short palindromic repeats and associated genes (CRISPR-Cas) and proto-spacers were predicted using CRISPRCasFinder (https://crisprcas.i2bc.paris-saclay.fr/CrisprCasFinder/Index, accessed on 15 September 2022) and CRISPRTarget (http://bioanalysis.otago.ac.nz/CRISPRTarget/crispr_analysis.html, accessed on 15 September 2022).

### 2.6. Accession of the Genome Sequences

The WGS data of *L. pneumophila* strain Corby (ICDC) have been deposited in the NCBI BioProject repository (accession number PRJNA904548) and the BioSample database (accession number SAMN31846690).

## 3. Results

### 3.1. Case Presentation

The patient was a 71-year-old woman who worked in an automobile factory assembling parts. The factory occasionally used air conditioners in summer, which were cleaned once per year. On 16 February 2022, she was admitted to a hospital in Beijing having had a cough and sputum for >5 months that was aggravated with shortness of breath for 1.5 months. She had visited the outpatient department 7 days before hospitalization. Test results showed that the left lung space was occupied with multiple inflammatory changes; there were metastatic lesions in the right lung. Chest enhanced computed tomography (CT) showed multiple infiltrates in both lungs. The volume of the left lung had reduced; the mediastinum was significantly left skewed, and the left lung apex showed effusion. After admission, the patient had fever and elevated white blood cell counts and inflammatory markers, and chest CT showed multiple infiltrates in both lungs and atelectasis of the left lung, which was considered to be caused by infection. The patient was given mask oxygen inhalation (5 L/min) and temporary anti-infective treatment with imipenem/cilastatin (0.5 g q6h); the oxygen pulse was maintained at an acceptable level. However, when the patient went to use the toilet without oxygen, the oxygen pulse decreased significantly; therefore, she was transferred to the intensive care unit and given mask oxygen inhalation (12 L/min). Next-generation sequencing (NGS) of peripheral blood, bronchoalveolar lavage fluid (BALF), and bronchial brushing specimens showed positive results for *L. pneumophila*. Transbronchial lung biopsy of the posterior basal segment of the left lower lobe identified a small number of abnormal glands, and heterotypic cells were seen growing along the alveolar wall in some of these areas. The disease was diagnosed as lung adenocarcinoma with *Legionella pneumophila* infection. After treatment with tigecycline, moxifloxacin, and ceftazidime, white blood cells and inflammatory markers were significantly decreased.

### 3.2. Antibiotic Susceptibility

Antibiotic sensitivity of the Corby (ICDC) clinical isolate was evaluated. The results showed that the strain was sensitive to nine antibiotics, including ciprofloxacin, levofloxacin, moxifloxacin, erythromycin, azithromycin, clarithromycin, rifampicin, tigecycline, and doxycycline (Table 1).

### 3.3. Cell Infection Assays

After 1.5 h of co-culture with mouse macrophages J774, the number of intracellularly growing *L. pneumophila* strain Corby (ICDC) and *L. pneumophila* strain Philadelphia 1 (JR32) was counted. The results showed Corby (ICDC) had stronger intracellular proliferation ability than JR32. At 48 and 72 h after infection, the number of intracellular bacteria was significantly higher in the Corby (ICDC) group than in the JR32 group (Figure 1).

### 3.4. General Features of the Legionella pneumophila Strain Corby (ICDC) Genome

It was found that *L. pneumophila* strain Corby (ICDC) has a circular genome of 3,452,077 bp and is composed of 29 genomic scaffolds (Figure 2). The GC content was 38.91%; 5 rRNAs and 42 tRNAs were present. The genome annotation identified 3201 protein-coding regions (CDS); the total length of coding genes was 3,046,908 bp, with an average length of 952 bp, accounting for 88.26% of the genome. No plasmids were identified in the genome. The main characteristics of the two Corby genomes analysed (such as genome length, G + C content and coding density) were found to be highly conserved (Table 2). 

### 3.5. Core–Pan Analysis and Phylogenetic Tree Construction

Core genes were defined as orthologous genes found in all samples. The genes excluding the core genes are collectively known as dispensable genes, and all dispensable genes combined with the core genes are collectively known as pan genes. Specific genes were unique to each sample and were not duplicated with other sample genes. Gene families in the pan-genome represent the capacity to accommodate genetic determinants, whereas gene families in the core genome are generally associated with bacterial replication, translation, and maintenance of cellular homeostasis. Corby (ICDC) was compared with 25 *L. pneumophila* reference strains (serotype 1) for the core–pan analysis, which identified 5771 pan genes and 2029 core genes. Corby (ICDC) contained 158 specific genes, and Corby contained 81 specific genes. The number of specific genes in the strains Alcoy, Paris, Lens, Philadelphia-1, and 130 b were 56, 27, 116, 8, and 154, respectively (Figure 3). The results further indicated that there were evolutionary differences between Corby (ICDC) and the 25 reference strains, and the significantly higher number of specific genes may be closely related to the dominant growth of Corby (ICDC) in complex external environments and may affect the pathogenicity and virulence of the strain. 

A phylogenetic tree using the NJ method showed that 26 *L. pneumophila* (serogroup 1) strains were divided into two major clades. The strains Lens, Paris, Philadelphia-1, Alcoy and Corby, which caused large outbreaks of *Legionella*, were all concentrated in branch A, and Corby (ICDC) belonged to the same branch as Alcoy and Corby, which were more closely related to Corby (Figure 4).

### 3.6. Multiple Genome Alignment

Genomic collinearity was further explored by comparing the complete genome of the strains *L. pneumophila* Corby and Corby (ICDC). We observed that Corby (ICDC) and Corby had 29 collinearity regions in their genomes and contained numerous genomic rearrangements and inversions (Figure 5). 

### 3.7. CRISPR-Cas

The CRISPR-Cas is an acquired immune system widely present in the genomes of prokaryotes such as bacteria and archaea, which mainly consists of CRISPR arrays and Cas genes [14,15]. The main function of CRISPR-Cas is to cut nucleic acids containing specific sequences, thereby providing the host with the ability to defend against invasion by foreign nucleic acids [16]. The tool CRISPRCasFinder identified the type IF system in Corby (ICDC), comprising six sets of *Cas* genes (*Cas1*, *Cas3-Cas2*, *Csy1*, *Csy2*, *Csy3*, and *Cas6*) and two sets of CRISPR arrays; the repeat sequences in the two CRISPR arrays were consistent, and both were 5′-GTTCACTGCCGCACAGGCAGCTTAGAAA-3′ (Figure 6). To deeply understand the CRISPR arrays, its spacers were analysed by CRISPRTarget (score ≥ 20). Eight spacers could be matched in CRISPR1, and three spacers could be matched in CRISPR2. We found that some spacers were associated with a single gene segment, and some spacers were associated with multiple gene segments. Exogenous nucleotides were mainly derived from bacteria, plasmids, phages, and viruses (Table 3). Through the analysis of known coding products, it was found that some spacer-matched gene encoded products closely related to microbial survival, such as capsid proteins; some were related to adaptation to the external environment, such as efflux pump transporters; whereas some of the matched genes encoded functional proteins, such as glycosyltransferases. There were also putative proteins whose functions currently remain unknown.

## 4. Discussion

In this paper, we report a case of *Legionella pneumophila* pneumonia in a patient with lung adenocarcinoma in China. The clinical manifestations included cough, expectoration, lung infiltration, fever, shortness of breath, and significantly decreased activity tolerance. For Legionnaires’ disease, tigecycline, moxifloxacin, and ceftazidime were administered according to the Chinese ‘guidelines for the diagnosis and treatment of community-acquired pneumonia’. Then, the white blood cells and inflammatory markers significantly decreased. In addition, pathogen isolation was performed by spreading the patient’s peripheral blood, sputum and bronchoalveolar lavage fluid on GVPC plates; the whole-genome sequence was obtained by WGS. The results showed that the isolate was *L. pneumophila*, with a high sequence similarity to the Corby isolate identified in England and was the second sequence of *L. pneumophila* strain Corby available worldwide and the first Corby strain isolated in China. 

The genomic and phenotypic characteristics of microorganisms provide a solid basis for understanding the characteristics of various strains and treating bacterial infections. In the present study, we evaluated the drug resistance and pathogenicity of Corby (ICDC) in vitro through antibiotic sensitivity and cytological tests. Macrolides and fluoroquinolones are the first-line antibiotics for the treatment of Legionnaires’ disease [17,18], of which erythromycin and azithromycin are the main representatives of macrolides [19,20]. However, when bacteria develop resistance to these antibiotics, it greatly increases the difficulty of clinical treatment. At present, 17% of the environmental and clinical *Legionella* isolates in China have been found to have azithromycin resistance, and the *lpeAB* efflux pump gene with azithromycin as the specific substrate exists in the genome [21,22]. In this study, the drug susceptibility of Corby (ICDC) to nine antibiotics belonging to the classes quinolones, macrolides, rifamycins, and tetracyclines was tested using the E test. The results showed that the MIC values of Corby (ICDC) to the nine antibiotics were significantly lower than the ECOFF value, and the routine clinical drugs are effective in the treatment of Legionnaires’ disease caused by Corby (ICDC). Moreover, the growth of *L. pneumophila* in macrophages is closely associated with human Legionnaires’ disease [23,24,25]. Cytological experiments showed that the strain had strong intracellular proliferation ability. Analysis of virulence factors revealed that the secretory systems of T2SS and T4BSS (Dot/Icm) were closely related to the pathogenicity of the strain. System T2SS plays an important role in the intracellular replication *L. pneumophila* infection of amoeba and human cells [26], whereas T4BSS of *L. pneumophila* is essential for growth and intracellular transport in human macrophages and amoeba [27,28]. In the cell, they form a unique replicating vacuole, called *Legionella*-containing vacuole (LCV), in which the strain escapes phagocytic lysosomal degradation, evades immune responses, and obtains nutrients essential for its growth and replication.

The WGS and comparative genomic analysis are the main methods for analysing strain biology, genetic evolutionary analysis, and horizontal gene transfer. The phylogenetic tree based on core genes showed that Corby (ICDC) was most closely related to the Corby strain isolated from England, and it was identified as serotype 1, which was the dominant serotype causing Legionnaires’ disease. Colinear alignment results revealed that Corby (ICDC) had a higher number of inverted and rearranged regions, the two isolates experienced a similar path during evolution, but Corby (ICDC) adapted to the complex changing environment by making partial changes to the genome under long-term selection pressure. However, it has been reported that *L. pneumophila* has plasticity and natural competence and possesses the complete recombination machinery and necessary characteristics for the integration of foreign DNA [29,30,31]. Therefore, it was not unexpected that a large number of genome rearrangements and inversions would be observed in Corby (ICDC) isolates.

The CRISPR-Cas is an adaptive immune system composed of CRISPR arrays and Cas genes, and the types IC, IF, and IIB have been reported in *Legionella* [32,33,34]. The CRISPR arrays contain short palindromic repeats (repeats) and spacers that separate the repeats. The repeats in CRISPR are almost identical, but the spacers are different and determine the specificity of the immune function of the CRISPR-Cas system. Previous studies have reported the CRISPR-Cas system is not present in Corby [32,35], but type IF was detected in the recently isolated Corby (ICDC) isolate, and two sets of CRISPR sequences were found to be present. Analysis of its spacers revealed abundant spacer-matched gene-encoded products, indicating that Corby (ICDC) was invaded by many kinds of bacteria, plasmids, viruses, and phages in the external environment; the major capsid protein encoded is a viral gene product, which has virus-specific antigenicity and stimulates the body to mount an antigenic viral immune response. In the spacer analysis, most of spacers matched the coding region or non-coding region sequence [36], which was consistent with the common spacer formation mechanism. However, some spacers matched multiple different gene segments, and it was speculated that these spacers are formed from the splicing and recombination of two or more gene segments. Few spacers simultaneously matched both coding and noncoding regions, and although this type of spacer is underrepresented, it may still represent a novel mechanism of spacer formation that merits further research to unravel the diversity of mechanisms of spacer formation [37]. Furthermore, most of the spacer sequences did not match the proto-spacer, and it is difficult to trace whether these spacers are derived from a plasmid, virus, or phage. There may be a several unexplored organisms in which rapid evolution may have led to numerous base mutations in the proto-spacer to escape spacer recognition.

## 5. Conclusions

Our findings highlight that Legionnaires’ disease caused by *Legionella pneumophila* strain Corby may be ignored in China owing to its sensitivity to the clinical routine antibiotics. The strong intracellular proliferation ability suggests that it can rapidly proliferate in human alveolar macrophages and cause Legionnaires’ disease and may have more undetected cases of infection in clinical practice. Furthermore, we need to pay attention to chronic lung diseases with *Legionella* infection. Enhanced long-term monitoring and investigation of *Legionella* in aquatic environments and patients with respiratory infections are necessary to prevent large-scale outbreaks of Legionnaires’ disease.

## Figures and Tables

**Figure 1 microorganisms-11-00204-f001:**
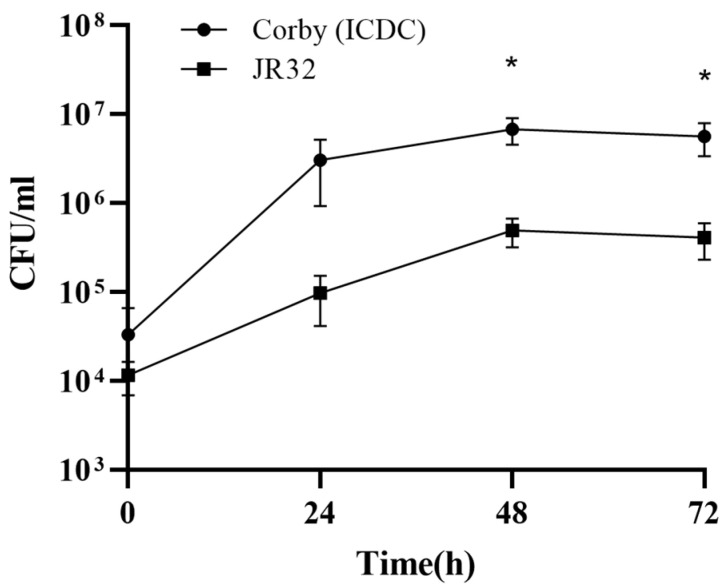
Comparison of intracellular proliferation in J774 mouse macrophages between *Legionella pneumophila* strain Corby (ICDC) and the control strain *L. pneumophila* Philadelphia 1 (JR32) (*, *p* < 0.05; Student’s *t* test).

**Figure 2 microorganisms-11-00204-f002:**
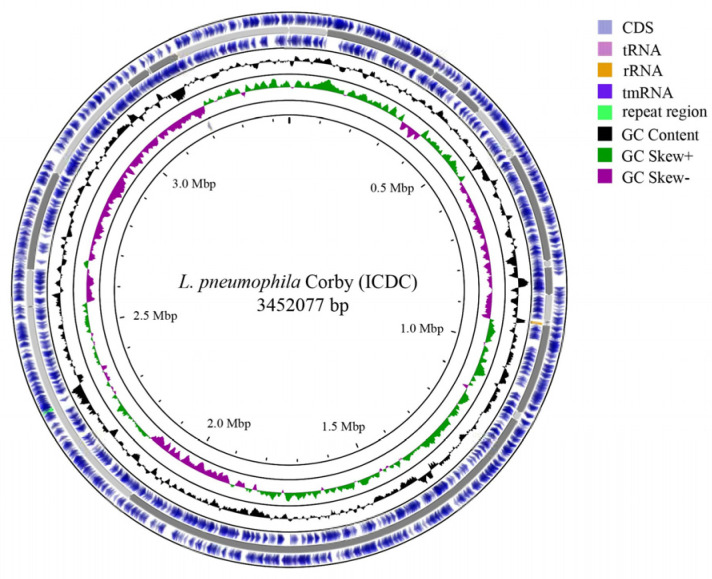
Genome visualization of *Legionella pneumophila* strain Corby (ICDC).

**Figure 3 microorganisms-11-00204-f003:**
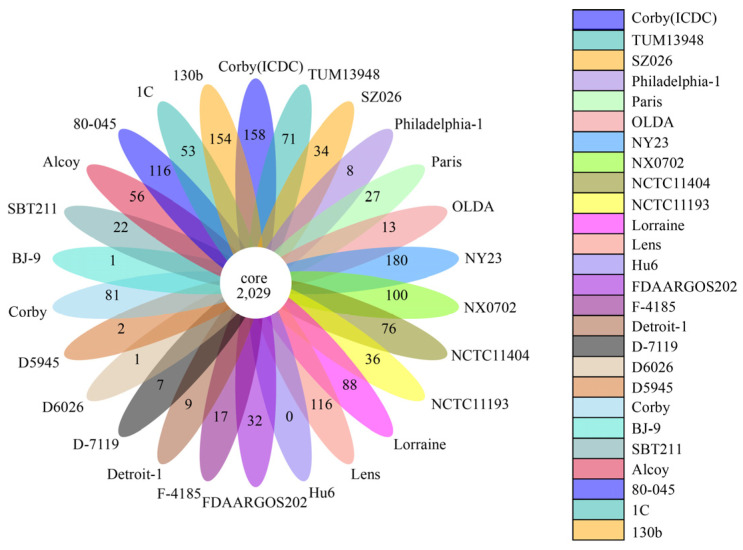
Core and specific genes of 26 genomes of serogroup 1 *Legionella pneumophila*. Each petal represents a strain genome. The number in the centre of the graph represents the core genes shared among the 26 strains. Numbers on each petal represent the genes specific for each strain.

**Figure 4 microorganisms-11-00204-f004:**
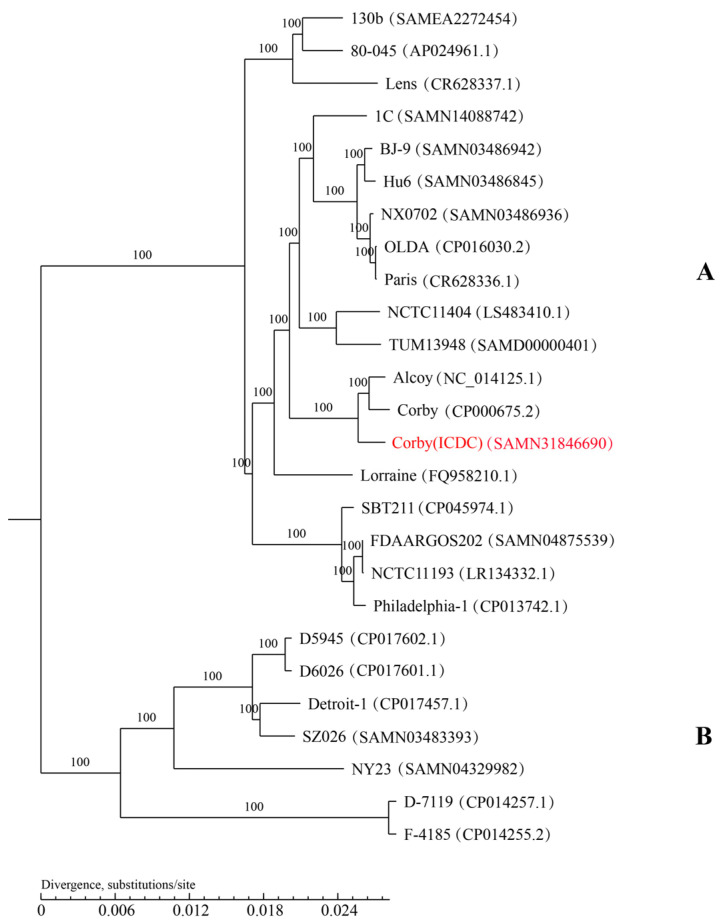
The phylogenetic tree is based on the core genes of *Legionella pneumophila*. The whole genome of *L. pneumophila* strain Corby (ICDC) (red label) isolated from a patient was compared to the genomes of serotype 1 *L. pneumophila*.

**Figure 5 microorganisms-11-00204-f005:**
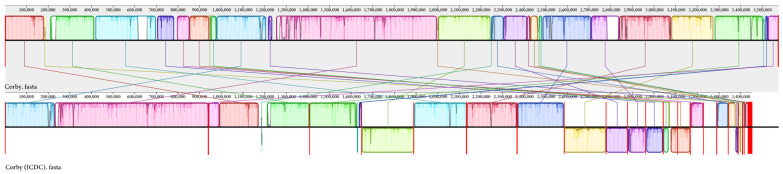
Comparison of the linear organization of *Legionella pneumophila* genomes. Local collinear blocks (LCBs) representing regions of sequence alignment are shown as coloured rectangles connected by straight lines. The LCBs placed above the line are in the same direction as the reference sequence; LCBs placed below the line are oriented in the opposite direction. Empty sections are regions of low sequence conservation.

**Figure 6 microorganisms-11-00204-f006:**
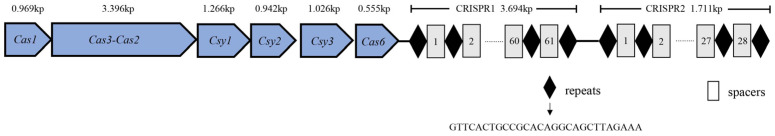
The type IF CRISPR-Cas locus of *Legionella pneumophila* strain Corby (ICDC).

**Table 1 microorganisms-11-00204-t001:** MIC values of antibiotics for the *Legionella pneumophila* Corby (ICDC) clinical isolate.

Antibiotic Class.	Drug	MIC Range (mg/L)	ECOFF (mg/L)	ATCC 33152	Corby (ICDC)
Quinolone	Ciprofloxacin	0.25–2	1.0	0.38 ^a^	0.25 ^a^
Quinolone	Levofloxacin	0.064–1	0.50	0.125 ^a^	0.064 ^a^
Quinolone	Moxifloxacin	0.25–1	1.0	0.75 ^a^	0.5 ^a^
Macrolide	Erythromycin	0.032–2	1.0	0.50 ^a^	0.19 ^a^
Macrolide	Azithromycin	0.038–8	1.0	0.25 ^a^	0.047 ^a^
Macrolide	Clarithromycin	0.064–1	0.50	0.19 ^a^	0.064 ^a^
Rifamycin	Rifampicin	0.004–0.032	0.032	<0.016 ^a^	<0.016 ^a^
Tetracycline	Tigecycline	1–16	16	0.5 ^a^	0.75 ^a^
Tetracycline	Doxycycline	1–8	8	4 ^a^	1.5 ^a^

MIC, minimum inhibitory concentration; ECOFF, epidemiological cut-off value, ^a^ MIC values within the range of susceptibility.

**Table 2 microorganisms-11-00204-t002:** Main features of *Legionella pneumophila* strain Corby (ICDC) and Corby genomes.

Features	Corby (ICDC)	Corby
Genome length (bp)	3,452,077	3,576,470
Serotype	1	1
G + C content (%)	38.91	38.48
Number of CDS genes	3201	3193
tRNA	42	44
16S/23S/5S	1/1/3	3/3/3
Average length of CDS (nt)	952	984.35
Number of plasmids	0	0

CDS, coding sequence.

**Table 3 microorganisms-11-00204-t003:** Coding products of partially spacer-matched genes in the CRISPR-Cas system of *Legionella pneumophila* strain Corby (ICDC).

CRISPR1				
Spacers	Coverage	Source of Proto-Spacers	Matching Category	Coding Product of Matching Genes
spacer12	27/32	bacteria	*Palaeococcus pacificus* DY20341 chromosome	Cobaltochelatase subunit *CobN*
	27/32	bacteria	*Thermococcus* sp. IOH2 chromosome	Cobaltochelatase subunit *CobN*
	26/32	bacteria	*Methanobrevibacter smithii*	NCR
	26/32	phage	*Arcanobacterium* phage vB-ApyS-JF1	Hypothetical protein and NCR
spacer26	27/32	virus	Gokushovirus isolate SH-CHD11	Putative *VP1*
	27/32	plasmid	*Citrobacter* sp. TSA-1 plasmid	Phage capsid protein
	26/32	virus	*Microviridae* sp. isolate 7408–1711	Similar to *VP1*
spacer29	26/32	bacteria	*Methanonatronarchaeum thermophilum* strain AMET1 AMET1_3	Glycosyltransferase family 4 protein
spacer41	26/32	plasmid	*Listeria monocytogenes* strain FDAARGOS_57 plasmid	Helix–turn–helix domain-containing protein
spacer43	26/32	bacteria	*Thermococcus radiotolerans* strain EJ2 chromosome	Flippase
spacer52	26/32	virus	Robinz microvirus RP_102	Major capsid protein
	26/32	virus	Gokushovirus WZ-2015a	*VP1*
spacer53	30/32	virus	Microviridae sp.	Major capsid protein
	27/32	virus	Gokushovirus WZ-2015a	*VP1*
	27/32	virus	Flumine microvirus	NCR
	27/32	virus	Robinz microvirus RP_84	Major capsid protein
	27/32	virus	Blackfly Microvirus SF02 isolate 174	Major capsid protein
	26/32	virus	Capybara microvirus Cap1_SP_192	Major capsid protein
	26/32	virus	Chimpanzee faeces-associated microphage 1 isolate CPNG_29298	Gene: major CP
spacer54	30/32	plasmid	*Legionella pneumophila* plasmid	Repeat region
	27/32	plasmid	*Burkholderia pseudomallei* strain 2008724860 plasmid p1	Chromate efflux transporter
NCR, non-coding region				
CRISPR2				
spacers	Coverage	Source of proto-spacers	Matching category	Coding product of matching genes
spacer2	27/32	plasmid	*Legionella pneumophila* subsp. *pneumophila* strain Allentown 1 (D-7475) plasmid	Repeat region
spacer17	26/32	phage	*Pseudomonas* phage PSA11	Hypothetical protein ORF028
		phage	Bacteriophage PA11	NCR
spacer23	26/32	phage	*Aeromonas* phage BUCT695	Scaffolding protein

NCR, non-coding region.

## Data Availability

The data presented in this study are openly available from the NCBI in the BioProject PRJNA904548 and the BioSample SAMN31846690.

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
