# Peer review of "Legionnaires’ Disease in China Caused by Legionella pneumophila Corby"

_microorganisms, 2023, doi:10.3390/microorganisms11010204_

Round 1
Reviewer 1 Report
This study aimed highlight L. pneumophila Corby as a public health threat in China. Moreover, owing to the plasticity of the Legionella genome and the fact that Legionella have an intact recombination machinery and the necessary features to integrate foreign DNA, it is essential to investigate its antibiotic resistance and virulence profile and its role in the pathogenesis of Legionnaires' disease and to perform a comparative genomic analysis to identify different features such as the CRISPR-Cas system and invasion by foreign genes
The results in tables and figures are well presented, so I have no objection to their presentation. The results are adequately discussed and compared with other works. The conclusion confirms the obtained results and the references inserted in the main document show a good connection between this investigation and other works.
The material and methods, as well as the results and discussion part, are understandable. The article is good from the grammatical and structural points of view, and from my perspective is acceptable for publication in this journal.
Minor Revision:
- The introduction is very short. A little more should be said about the L. pneumophila Corby bacterium.
- The introduction must clearly contain the objective of the work.
- The scientific name of the bacteria must always be in italics correct throughout the manuscript.
- Conclusion is short.
- Review the format of bibliographic references so that it is unified.
Best regards,
Reviewer 2 Report
The authors described that Legionella pneumophila Corby was first identified in China and infected patient causing Legionnaires' disease. The findings revealed the threat of L. pneumophila Corby to public health in China, and emphasized the importance of strengthening the detection of Legionella in aquatic environment and patients with respiratory tract infection. Authors also tested the sequence composition of the strain and its amplification ability in cells. The manuscript is well competed, and only some minor problems need to be modified.
Line 35-36. The statement about the mortality of Lp is easy to cause misunderstanding. It is suggested that the author revise this sentence as “the epidemic mortality due to the disease can reach 30%”, or specify the range of the mortality.
Line 45-47. This part of the author's statement is not concise, and please elaborate on the current research status of Legionella pneumophila Corby.
Line 62. Please supplement the full name of the abbreviation NGS.
Line 90. Please specify the culture conditions, plates, and culture medium used for J774 cells.
In part 3.2, authors listed 9 sensitive antibiotics of ICDC. Whether authors had detected other antibiotics at the same time, and whether there are antibiotics insensitive to the strain. If so, authors should list the results so that readers could better understand the sensitivity of this strain to antibiotics.
Line 148-150. “The findings indicated that the chief macrolides and quinolones used in the clinical treatment of Legionnaires' disease can effectively treat the infection caused by ICDC”. The interpretations of findings should be included in the discussion section and not in the results section, and the subsection is already included in the discussion. It is recommended to delete.
Line 229. The author mentioned that exogenous nucleotides mainly come from bacteria, plasmids and viruses, but according to Table 3, the source of proto-spacers also includes phage, which should be correctly expressed.
In line 60 and 248-249. In the materials and Methods section, it is mentioned that peripheral blood, sputum, and bronchoalveolar lavage fluid (BALF) of the patients were collected for pathogen isolation and culture, but in the discussion section, it is stated that sputum and bronchoalveolar lavage fluid (BALF) were used for pathogen isolation and culture. Please be consistent with the statements.
Line 317-319. Authors only tested the amplification ability of the strain in cells, however, the amplification ability in cells is not always positively related to the pathogenicity of Lp strains. Therefore, the expression of strain pathogenicity here lacks sufficient support.
Reviewer 3 Report
This case report provides a description of a very detailed investigation into legionellosis and the first recorded instance with this strain in China. The range of assays undertaken to establish the identity and characteristics of the patient isolate are well described. The analytical approaches described provide a good template for other case investigations.
There are just a small number of minor editorial changes needed and I think on page 10 line 276 it should read 'Legionella containing vacuoles (LCV)'?
My one main question. The authors may consider this outside the scope of the case report but I'm curious about how a Legionella strain originating from an industrial outbreak some years ago in central England can turn up in Beijing. Any hypothesis? Also, assuming that the automobile factory was the source (it doesn't say if this was established) are the authors aware of any interventions put in place there to prevent others being infected?
